# Numerical modelling framework for assessing dune effectiveness against coastal inundation

**Italo dos Reis Lopes**[1,2]**, Ivan Federico**[2]**, Michalis Vousdoukas**[3]**, Luisa Perini**[4]**, Salvatore Causio**[2]**, Giovanni Coppini**[2]**, Maurilio Milella**[5]**, Nadia Pinardi**[1,2]**, and Lorenzo Mentaschi**[1,2]

[1]Dipartimento di Fisica e Astronomia "Augusto Righi" (DIFA) – Alma Mater Studiorum
Università di Bologna (UNIBO), Bologna, Italy
[2]CMCC Foundation – Euro-Mediterranean Center on Climate Change, Italy
[3]University of the Aegean, Department of Marine Sciences, Mytilene, Greece
[4]Area Geologia, Suoli e Sismica – Regione Emilia-Romagna, Bologna, 40127, Italy
[5]Environmental Surveys S.r.l. (ENSU), Spin off University of Bari "Aldo Moro", Bari, 70125, Italy

**Correspondence:** Italo dos Reis Lopes (italo.dosreislopes2@unibo.it)

**Abstract.** Coastal inundation threatens both economic assets and human lives, yet accurate flood mapping remains limited by gaps in data availability and model capabilities. In this study, we enhanced the LISFLOOD-FP model to simulate coastal floods by incorporating wave setup, swash dynamics, and interactions with protective infrastructure such as temporary dunes. We applied this approach to Cesenatico, Italy, where seasonal dunes serve as winter coastal defenses, analyzing two contrasting storm events with observational data for validation: the 2015 Saint Agatha Storm, which breached the dunes causing extensive inland flooding, and the 2022 Denise Storm, where intact dunes successfully prevented inundation. Our results demonstrate that dunes effectively mitigate flooding when intact, but failure of even small sections can trigger widespread inundation, highlighting the critical need for optimized design. This work advances the development of coastal digital twins by introducing a computationally efficient representation of essential physical processes – swash-related erosion of dune stability and swash contribution to flood volumes through an overwash efficiency parameter – enabling practical risk assessment and infrastructure planning in vulnerable coastal regions.

## 1   Introduction

Floods are substantial environmental hazards that impact global populations and present significant socio-economic challenges (UNODRR, 2020). In Europe, climate-related economic losses between 1980 and 2020 are estimated in EUR 450 to 520 billion, with hydrological events being the most impactful (44 %) (European Environmental Agency (EEA), 2024). Projected scenarios indicate that coastal flood-related damage could reach up to EUR 1 trillion annually by 2100, due to the ongoing climate changes and the related Sea-Level Rise (SLR). Climate Change, urbanization and migration into coastal areas contribute for a rise in coastal exposure (IPCC, 2018, 2021) emphasizing the critical importance of accurate representation and effective management of coastal flood events for risk prevention.

Flood numerical modeling techniques vary widely from simple bathtub models (Didier et al., 2019; Williams and Lück-Vogel, 2020), which tend to overestimate flooding, to comprehensive representations of hydro-morphodynamical processes (Vousdoukas, 2012; Wilmink et al., 2023), which require extensive data inputs and computational resources. Intermediate complexity models, solving the shallow water equations for floodplains, such as LISFLOOD-FP (Bates and De Roo, 2000; Bates et al., 2010; Shaw et al., 2021), offer a good balance between accuracy and computational efficiency. Initially developed to extend the LISFLOOD model

for river channels and floodplain inundations, LISFLOOD-FP has proven to have skills comparable to more complex hydrological inundation models while using lower computational resources (Smith et al., 2012; Vousdoukas et al., 2016; Bessar et al., 2021). In the European Flood Awareness System (EFAS) project, LISFLOOD-FP is used on a large scale to create datasets of river flood hazard maps by using hydrological data over various return periods to generate flood scenarios (Dottori et al., 2022).

LISFLOOD-FP is also a popular choice for coastal flood modelling. Indeed, the European Coastal Flood Awareness System (ECFAS) project (Le Gal et al., 2023; Irazoqui Apecechea et al., 2023), relies on LISFLOOD-FP to simulate inundations in the coastal area. The ECFAS project aims to improve flood awareness and preparedness along the European coastline, focusing on the significant economic impacts of coastal flooding. Le Gal et al. (2023) developed comprehensive flood maps generated by LISFLOOD-FP for different coastal sectors on the European coasts, considering synthetic scenarios that provide general insights into flood hazard assessment across Europe. However, local-scale coastal studies highlight the limitations in our capabilities to predict this phenomenon. To start, accurate coastal flood modelling requires accurate data of Total Water Level (TWL), which consists of Sea Surface Height (SSH) and wave components, to accurately represent flood extent (Zhang and Najafi, 2020; Carneiro-Barros et al., 2023). This implies that LISFLOOD-FP offshore lateral open boundary conditions must include information as accurate as possible on sea-level and waves.

Another modelling limitation discussed by Dottori et al. (2022) and Carneiro-Barros et al. (2023) is the non-inclusion of defenses in the simulations. Coastal defenses encompass a range of structures, including hard engineering solutions, Nature-Based Solutions (NBS), and hybrid forms that vary in structural complexity and interaction with storm events (Almarshed et al., 2020). Incorporating these defensive structures into numerical models poses a challenge, as accurate representation requires detailed information on defense geometries and potential modifications in response to flooding. Coastal dunes, a form of NBS commonly found on sandy shorelines, provide effective protection against storm surge-induced flooding by acting as barriers near the beach interface in backshore (Wijnberg et al., 2021; Singhvi et al., 2022). However, the numerical modeling of coastal dune erosion presents complex challenges due to the combined effects of storm surges and wave overwash (van Wiechen et al., 2023).

Coastal dune erosion refers to the landward retreat of sandy beaches and dune systems as a result of storm-induced wave action and elevated water levels. The extent of this erosion can be described using an erosion hazard scale (Leaman et al., 2021) based on the degree of horizontal recession experienced during a storm. At the lowest level, minor beach narrowing occurs when the beach width is reduced but the dune system remains unaffected. As erosion intensifies, substantial beach narrowing takes place, where the dune system is still intact but becomes more vulnerable to damage from subsequent storms. More severe conditions lead to dune face erosion, in which erosion progresses landward from the dune toe but does not yet reach the crest. Under the most extreme circumstances, dune retreat occurs, where significant erosion impacts and undermines the landward side of the dune crest, leading to a loss of dune volume and a reduction in the coastal buffer that protects inland areas from storm surges and flooding.

The Italian region of Emilia-Romagna (ER) is an example of a low-lying area vulnerable to coastal flood events usually associated with a combined effect of surge, tides, and waves. Several studies account for the economic losses (Carisi et al., 2018; Armaroli et al., 2019) and hazard assessment impact (Ciavola et al., 2007; Martinelli et al., 2010; Armaroli et al., 2012). To reduce the hazard, temporary dunes have been built as coastal defenses in November and maintained during the Winter until April. Harley and Ciavola (2013) conducted risk assessments studies related to these seasonal dunes and propose a GIS based methodology for engineering the dunes' geometry. In flood modeling, these dunes present two main challenges: (a) accurately representing their elevation and spatial distribution in the topographic data, and (b) understanding and modeling their potential responses, including structural failures, during extreme events.

Enhancing our coastal flood modeling capabilities is a critical step toward developing a digital twin for sustainable coastal management. Digital twins are advanced virtual replicas of the physical systems, enabling scenario simulations and exploration while using observational data to continuously refine and calibrate models. Although initially popular in the industrial sector, digital twin technology has recently been adopted for environmental applications across Europe (Nativi et al., 2020). As an example, in Emilia-Romagna Pillai et al. (2022) explores this concept, highlighting the potential of digital twins to improve understanding of wave attenuation through Nature-Based Solutions such as seagrass in the region offshore the coasts. The development of an accurate coastal flood numerical model would enable the exploration of what-if scenarios with the aim of developing an optimal layout of coastal defenses.

To address these gaps, here we introduce novel approaches tailored to parametrize the dynamics of coastal protections and wave component within the LISFLOOD-FP model. Specifically, we incorporate dune structures and a Failure Water Depth (FWD) threshold to simulate their collapse and introduce the effect of wave swash on erosion and on flood water supply.

Our developments were tested by carrying out simulations for two flood events in Cesenatico (Emilia-Romagna, ER) for which observed flood maps are available, provided by the Geological, Soil and Seismic Area of the ER region. The first is the Saint Agatha Storm, which occurred in February 2015. During this event, a significant portion of the artificial dunes

along the coast failed resulting in a major flood (Perini et al., 2015). The second event is the Denise Storm, which took place in November 2022, causing a combination of surge, tides and waves that led to floods in part of the region, however in this case the artificial dunes provided an effective protection in sarge swaths of the coasts.

In Sect. 2 the numerical modelling improvements for the Total Water Level are described, including the Swash boundary forcing, the dune failure assumptions and all the input data used. In Sect. 3 we show the numerical simulations with and without dunes during the two flood events and Sect. 4 concludes with a discussion.

## 2 Methods and data

### 2.1 Contribution of waves to coastal water levels

Waves contribute in a complex way to Total Water Levels (TWL), which is defined as the combination of tides, surge, and wave runup (composed by the wave setup and swash). The wave setup associated with the wave dissipation and the related decrease in radiation stress, provide a neat contribution to the coastal sea-level (e.g. Melet et al., 2018). The swash is the intermittent water wash-up on the beach as the waves finally break. Although the swash has no effect on the mean sea-level, it contributes to coastal hazard and inundation in at least 2 ways: (1) in extreme conditions the runup is a major driver of erosion, possibly contributing to the collapse of coastal defenses like sandy dunes. (2) A continuous overwash can provide a substantial water supply for coastal inundation compared to mean coastal sea-level.

LISFLOOD-FP model domain starts from the coastline that is considered the open boundary condition for Total Water Level (TWL). It then extends on the land as far as necessary. Here we improved the parameterization of the contribution of waves introducing the swash ($S$) in the TWL, estimated consistently with existing literature (e.g. Stockdon et al., 2006) as the sum of the water level plus half the swash

$$\text{TWL} = \text{WL} + S/2 \tag{1}$$

where the term WL is assumed to contain all the contribution to mean coastal sea-level, including the wave setup. TWL is then used in the model to parameterize the possible collapse of coastal defenses (see Sect. 2.2). Furthermore, we considered the contribution of waves to water supply for inundation by introducing the concept of Supply Total Water Level (STWL) (Fig. 1), given by:

$$\text{STWL} = \text{WL} + \alpha/2\,S \tag{2}$$

TS1 where $\alpha \in [0, 1]$ is a calibration factor which represents the overwash efficiency. A value of $\alpha = 0$ implies that the contribution of waves is limited to the wave setup, which

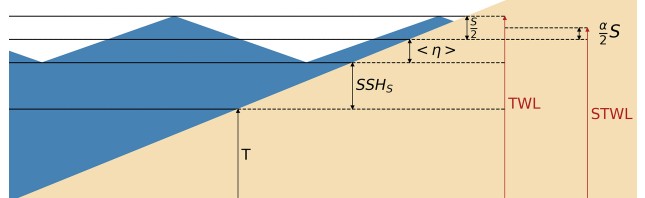

**Figure 1.** Schematics of the ocean components for the water level and the water supply associated with the swash where $T$ is the tide component, $\text{SSH}_\text{s}$ is the Sea Surface Heigh due to surge, $\langle n \rangle$ is the wave setup, $S$ is the swash, $\alpha$ is the overwash efficiency, TWL is the Total Water Level and STWL is the Supply Total Water Level.

could lead to an underestimation of the water supply driving the flood, as the overwash effect would be ignored. Conversely, $\alpha = 1$ assumes that the swash continuously contributes to the water supply, disregarding the intermittent nature of overwash and potentially resulting in an overestimation of the water supply.

In our model configuration, the WL is considered as the contribution of the Sea-level component due to atmospheric forcing (surge) ($\text{SSH}_\text{s}$), Tides ($T$) and the wave setup $\langle\eta\rangle$

$$\text{WL} = \text{SSH}_\text{s} + T + \langle\eta\rangle \tag{3}$$

The wave-induced contributions were estimated using the approach suggested by Stockdon et al. (2006), which states that the wave setup and swash for waves perpendicular to the coast is given by

$$\langle\eta\rangle = 0.35\beta_\text{f}\,(\text{HL})^{1/2} \tag{4}$$

$$S_0 = \left\lfloor \text{HL}\left(0.563\beta_\text{f}^2 + 0.004\right)\right\rfloor^{1/2} \tag{5}$$

where $\langle\eta_0\rangle$ and $S_0$ are the wave setup and swash assuming a wave propagation perpendicular to the shore, $\beta_\text{f}$ is the beach face-slope which is the inclination of the beach portion between the high tide and low tide lines, $H$ is the Significant Wave Height, $L$ is the mean wavelength. According to Stockdon et al. (2006), the values of $\langle\eta\rangle$ and $S_0$ are further multiplied by a 1.1 factor, which is the regression coefficient between $\langle\eta_0\rangle + S_0/2$ and the observed wave runup. Considering waves directed with an angle $\theta$ relative to the shoreline, the wave setup and swash set as boundary condition in LISFLOOD-FP in this study is given by

$$\langle\eta\rangle = 1.1 \cdot \max\,(\sin\theta, 0)\,\langle\eta_0\rangle \tag{6}$$

$$S = 1.1 \cdot \max\,(\sin\theta, 0)\,S_0 \tag{7}$$

### 2.2 Coastal defense structure modelling

The approach proposed in this study draws inspiration from Shustikova et al. (2020), who developed a methodology for the representation of levees and their breaching processes.

It consists of adding protective, sometimes non-permanent features such as dunes to the Digital Terrain Model (DTM), allowing the model to reproduce their effect in blocking the water flow (Fig. 2). For each time step, the Total Water Level (TWL) in the vicinity of the dune is compared with its Failure Water Depth (FWD), that is a threshold for dune erosion (e.g. van Rijn, 2009). When the FWD is exceeded, the dune is entirely eroded and removed from the terrain. For the sides of the dune facing the offshore, the FWD is compared to the full TWL, and not the STWL, as the swash plays a prominent role in dune erosion.

## 2.3    Model setup and input data validation

Among the numerical schemes available in LISFLOOD-FP we selected the acceleration scheme, which offers a trade-off between accuracy and computational parsimony. The model requires lateral boundary conditions for TWL at the coastline, DTM data and dune's position/geometry. The TWL, defined in Eq. (1), is the combination of sea-level and wave data from a large-scale model. These data are provided by the hindcast of Mentaschi et al. (2023), which attains a resolution of 2–4 km along the global coast.

Prior to carrying out simulations, we validated the hindcast data for the years 2015 and 2022. We compared the modeled Sea Surface Height (SSH) from Mentaschi et al. (2023) with the data from the tidal station in Porto Corsini, provided by the Istituto Superiore per la Protezione e la Ricerca Ambientale (ISPRA), and the Significant Wave Height (SWH) with the data of the Nausicaa buoy, provided by the Agenzia Regionale per la Prevenzione, Ambiente Energia dell'Emilia-Romagna (ARPAE) (Fig. 3).

To compare Porto Corsini tide gauge data with the model simulation of Mentaschi et al. (2023), which does not include tidal information, the Porto Corsini tide gauge data had to be filtered from the tidal signal. A quasi-daily residual tidal-like signal is still evident in the time series of Fig. 4, likely due to the seiches, which in the Adriatic Sea have a period of roughly 22 h (Medvedev et al., 2020). Visually the comparison is very good, quantitatively the correlation between the hindcast and observed data is 95 %, and the RMSE is 0.02 m for 2015. In 2022, the correlation is 85 %, and the RMSE is 0.02 m.

Regarding wave data, we limited our comparison of Significant Wave Height (SWH) to the year 2015, as the data for 2022 were unavailable from the Nausicaa buoy. The correlation is 97 %, with a negative significant wave height BIAS of $-0.04$ m and an RMSE of 0.02 m (Fig. 5). These results, show that we can reasonably assume the hindcast provides a reliable representation of the study area and is suitable for use as input data in the model.

The DTM was provided by the Geological, Soil and Seismic Area of the ER region with a spatial resolution of 5 m, referenced to the WGS84/UTM Zone 32N coordinate system (EPSG:32632) and an acquisition date of 2009. A coastline mapping was carried out to provide boundary points coordinates in the sea/land interface. The coastline is determined by analyzing the DTM, identifying the zero-crossing, and designating the first positive point as its location. The model was set on a domain covering the area of Cesenatico with a resolution of 50 m. The resulting grid has a size of $150 \times 121$ grid cells.

For the simulations in Cesenatico (ER), a mapping of the seasonal dunes in the area was carried out using high-resolution Google satellite imagery acquired in March 2015. Since these images were taken after the storm event, only the locations where dunes withstand the storm or had been reformed could be clearly identified. The mapping focused on delineating the spatial position of dune crests through visual interpretation of the dune ridges. Only the geolocation points were incorporated into the model, while dune geometry (width) was constrained by the 50 m grid resolution. As detailed information on dune morphology for 2015 and 2022 was unavailable, we sought a configuration capable of accurately reproducing both events. The dune height was assigned a uniform value corresponding to a Failure Water Depth (FWD) of 1.4 m to all grid cells, based on a sensitivity analysis. However, the model structure allows the assignment of different FWD or dune height values for each grid cell, enabling future applications to incorporate spatial variability when more detailed morphological data become available.

The boundary conditions were generated using $SSH_s$ and wave components from the hindcast's closest node, the tide component T from Porto Corsini and the coastline angle. To compute the TWL written in Eq. (1), the beach-face slope for the study area was set to $\beta_f = 5 \%$ according to Ciavola et al. (2006) and the overwash efficiency was set to $\alpha = 0.25$ based on geometrical considerations, and approximating the waves as triangular (Fig. 1). By approximating an individual wave as a triangular shape, the ratio of its mean height (representing the effective water volume contributing to overwash) to its maximum height is 0.5. However, only the uprush portion of the wave contributes to overwash transport, while the backwash is typically lost seaward. Assuming an approximately equal division between uprush and backwash, the effective fraction becomes $0.5 \times 0.5 = 0.25$. Hence, $\alpha = 0.25$ represents the fraction of the incident wave height that contributes effectively to overwash transport under this simplified geometrical assumption.

Although this argument is based on a symmetric triangular waveform, the same reasoning applies to asymmetric, sawtooth-like waves characterized by a steep uprush and a more gradual backwash – waveforms commonly observed in the surf and swash zones and frequently adopted as first-order approximations in coastal engineering (Suntoyo et al., 2008; Grasso et al., 2011; Bonneton, 2023). In such cases, the geometric asymmetry alters the relative duration of the uprush and backwash phases but does not fundamentally change the proportional relationship between total wave height and the effective uprush volume contributing to overwash. Therefore,

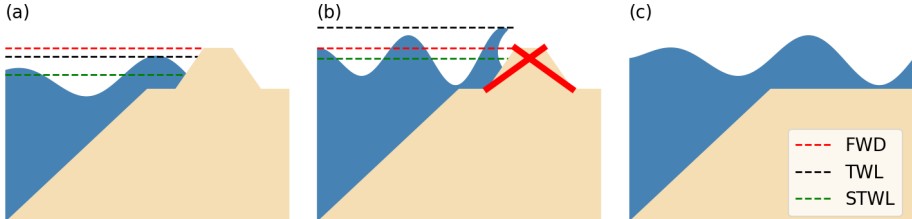

**Figure 2.** Schematic representation of coastal protections in LISFLOOD-FP. **(a)** protective action when TWL < FWD. **(b)** failure for TWL > FWD. **(c)** free flood propagation upon protection failure. Red line represents the FWD, black line the TWL and green line the STWL.

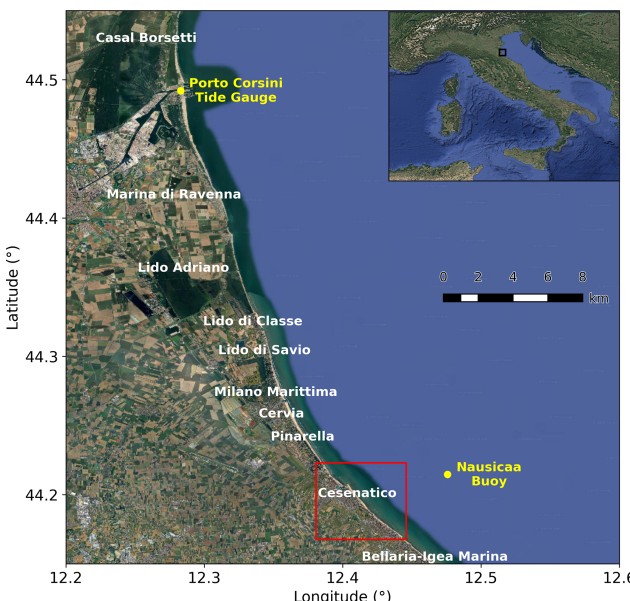

**Figure 3.** Emilia-Romagna's coast in the northeast of Italy. Yellow dots represent Porto Corsini's tide gauge (north) and Nausicaa's wave buoy (south). Red rectangle represents the modeled area in the town of Cesenatico. ©Google Maps.

the chosen value of $\alpha = 0.25$ remains a reasonable, physically consistent approximation for both symmetric and asymmetric (saw-tooth) wave shapes.

With this configuration, 9 different boundary conditions points are obtained along the coast (Fig. 6a). The boundary conditions for 2015 and 2022 for the different points are, then, presented in Figs. 6 and 7, respectively.

## 2.4 Simulation experiments and validation

Table 1 contains the description of the numerical experiments. The simulations were carried out for two specific flood events: the storm Agatha of 2015 (from 2 February to 6 February 2015) and the storm Denise of 2022 (from 22 November to 23 November 2022). To understand the dune's contribution to the flood, for each event, 2 simulations were carried out: one without (E2015 and E2022) and one with (E2015D and E2022D) dunes.

Moreover, to understand the waves' contribution to the flood, simulations were carried out by neglecting the contribution of the swash (E2015DWL and E2022DWL) and assuming that waves fully contribute to both dune failure and water supply, by setting a boundary condition equal to TWL (E2015DTWL and E2022DTWL).

Furthermore, we estimated how uncertainty in the DTM propagates in the results of the flood model. Duo et al. (2018) quantified the uncertainty in beach profiles by comparing the measurements of two different instruments in post-storm conditions after the 2015 event. The Root Mean Square Error (RMSE) between these instruments was found to be 0.12–0.14 m. Assuming our DTM exhibits a similar range of uncertainty, we conducted additional simulations by adding or subtracting a confidence value of 0.07 m to the DTM. These simulations were performed both with dunes (E2015D+, E2015D−, E2022D+, E2022D−) and without dunes (E2015+, E2015−, E2022+, E2022−), where the ± suffix indicates the addition or subtraction of the confidence value. TS2 We then estimated the uncertainty associated with dunes (UDUNE2015 and UDUNE2022), waves (UWAVE2015 and UWAVE2022) and DTM (UDTM2015 and UDTM2022) as the difference between simulations (Table 2).

The maximum flood extension simulated by LISFLOOD-FP was compared with the observations for each event. The grid points where the model reproduced water levels lower than 10 cm were neglected. For the comparison, the set of skill indicators suggested by Vousdoukas et al. (2016) was used:

– The BIAS is defined as the percentage ratio between predicted and observed area, and values lower (higher) than 100 % indicate an underestimation (overestimation) of the flooded area. It is given by

$$\text{BIAS} = 100 \times \frac{F_\text{m}}{F_\text{o}} \qquad (8)$$

where $F_\text{m}$ and $F_\text{o}$ are the extent of the modelled and observed flooded areas.

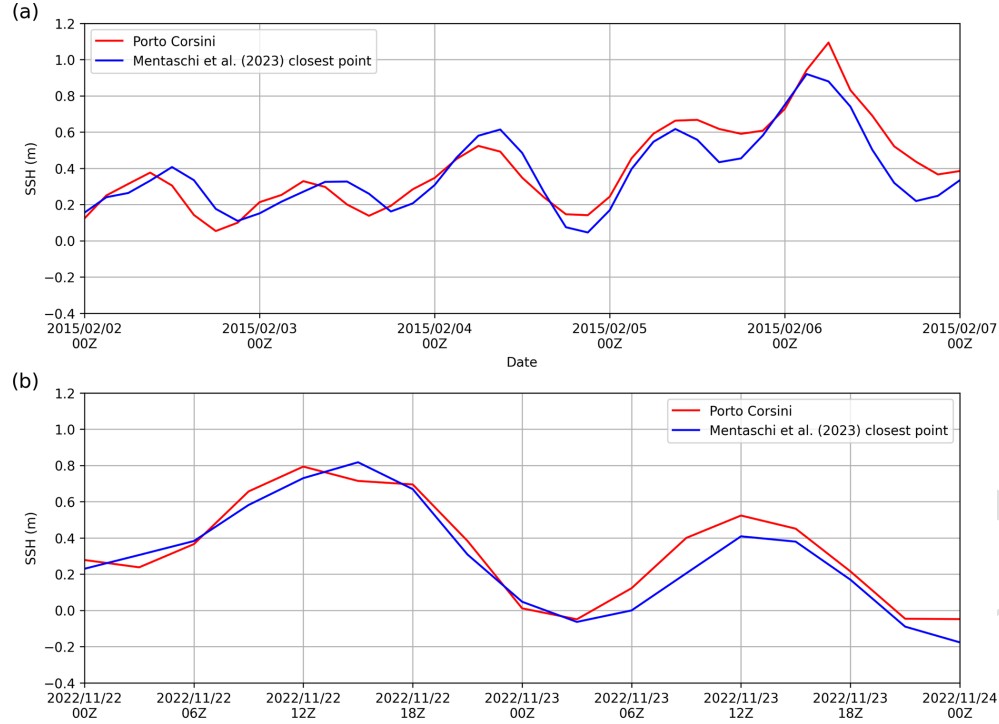

**Figure 4.** Comparison between the data of storm surge from Mentaschi et al. (2023) (blue line), with the filtered SSH data observed at the Porto Corsini station (red line) for the 2015 event **(a)** and 2022 event **(b)**.

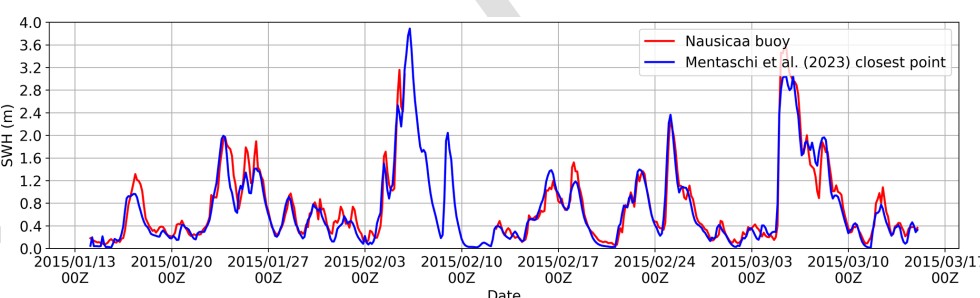

**Figure 5.** Timeseries of SWH for the Nausicaa buoy (red line) and from Mentaschi et al. (2023) (blue line) for the 3-month event centered comparison in 2015.

– The false alarm $F$ is the percentage ratios between wrongly inundated pixels and the observed ones. High values of this indicator indicate a high amount of wrongly inundated areas.

$$F = 100 \times \frac{F_m \neg F_o}{F_o} \qquad (9)$$

where $F_m \neg F_o$ indicates the extent of the area flooded in the model but not in the observations.

– The hit ratio ($H$) provides the opposite information with respect to $F$, which is an indication on the degree of agreement between the correctly modelled and the observed flooded areas. It is defined as the percentage ratio between the intersection of the modelled/observed

flooded areas ($F_m \cap F_o$) and the observed flooded area.

$$H = 100 \times \frac{F_m \cap F_o}{F_o} \qquad (10)$$

– The critical success index ($C$) is a renormalization of $H$ that results in the penalization of the indicator in case of high false alarm. It is defined as the percentage ratio between the intersection of the modelled/observed flooded areas ($F_m \cap F_o$) and the union of the two ($F_m \cup F_o$).

$$C = 100 \times \frac{F_m \cap F_o}{F_m \cup F_o} \qquad (11)$$

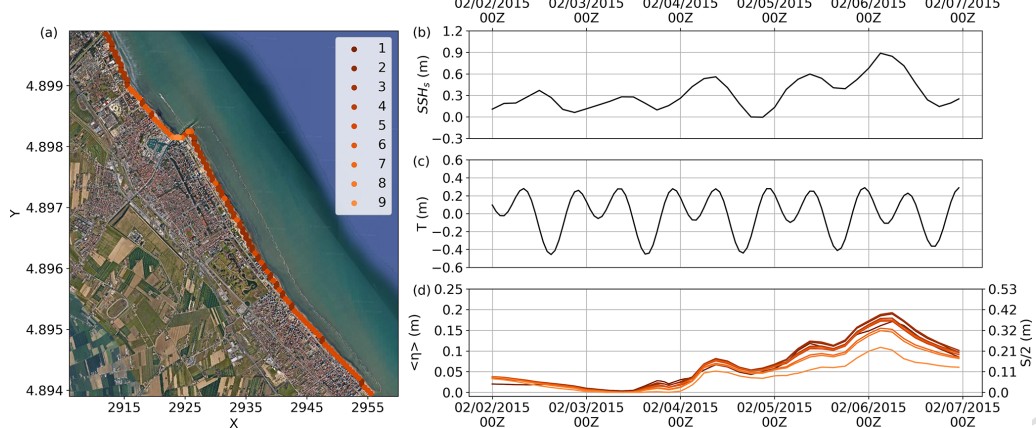

**Figure 6.** Boundary condition points associated with coastline angle along the Cesenatico area **(a)**. Darker (lighter) colors indicate a more meridional (zonal) coastline orientation (from Google Maps). Panels **(b)**–**(d)** show the boundary conditions for SSHs (m) **(b)**, $T$ (m) **(c)**, and both $\langle\eta\rangle$ (m) and $S/2$ (m) **(d)** during the 2015 event. In panel **(d)**, $\langle\eta\rangle$ and $S$ are proportional; the left $y$-axis corresponds to $\langle\eta\rangle$, while the right $y$-axis represents $S/2$, allowing both quantities to be conveyed by a single curve. ©Google Maps.

**Table 1.** Simulations configurations using different dunes, lateral boundary conditions, DTM offset and simulation period. TS3

| Simulations | Dunes | Dune's failure condition | Boundary condition | DTM offset | Simulation period |
|---|---|---|---|---|---|
| E2015 | No | None | | 0 m | 2 Feb 2015 |
| E2015D | | | STWL | 0 m | 00:00:00 to |
| E2015D+ | Yes | TWL | | +0.07 m | 6 Feb 2015 |
| E2015D− | | | | −0.07 m | 23:00:00 |
| E2015DWL | Yes | WL | WL | 0 m | |
| E2015DTWL | | TWL | TWL | | |
| E2022 | No | None | | 0 m | 23 Nov 2022 |
| E2022D | | | STWL | 0 m | 23:00:00 to |
| E2022D+ | Yes | TWL | | +0.07 m | 6 Feb 2015 |
| E2022D− | | | | −0.07 m | 23:00:00 |
| E2022DWL | Yes | WL | WL | 0 m | |
| E2022DTWL | | TWL | TWL | | |

**Table 2.** Uncertainties definition as the difference between simulations.

| Uncertainty | Simulations |
|---|---|
| UDUNE2015 | (E2015D) – (E2015) |
| UDUNE2022 | (E2022D) – (E2022) |
| UDTM2015 | (E2015D+) – (E2015D−) |
| UDTM2022 | (E2022D+) – (E2022D−) |
| UWAVE2015 | (E2015DTWL) – (E2015DWL) |
| UWAVE2022 | (E2022DTWL) – (E2022DWL) |

## 3 Results

### 3.1 Effects of dunes in inundation

In Fig. 8, the maximum flooded area roughly corresponds with the observed one (cyan line) for the 2015 event. The simulations with and without dunes (E2015D and E2015) are in substantial agreement and reproduce a major coastal flood. However, the simulations show overestimation with a broader flood inland for the whole area except in the south. For E2015 (Fig. 8a), the maximum water depth presents lower values in the northern and southern portions of the domain (0.7 m, in general) and higher values in the center (> 1.0 m), with a maximum of 1.18 m. In E2015D (Fig. 8b),

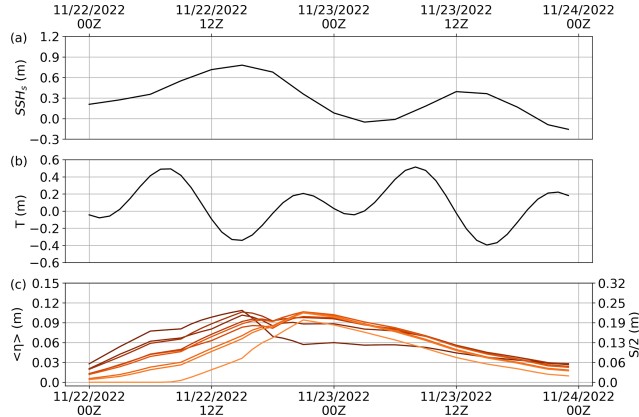

**Figure 7.** Boundary conditions for SSHs (m) **(a)**, $T$ (m) **(b)**, and both $\langle\eta\rangle$ (m) and $S/2$ (m) **(c)** during the 2022 event. In panel **(c)**, $\langle\eta\rangle$ and $S$ are proportional; the left $y$-axis corresponds to $\langle\eta\rangle$, while the right $y$-axis represents $S/2$, allowing both quantities to be conveyed by a single curve. Darker (lighter) colors indicate a more meridional (zonal) coastline orientation.

**Table 3.** Evaluation metrics for simulations E2015, E2015D, E2022 and E2022D.

| Metrics | E2015 | E2015D | E2022 | E2022D |
|---|---|---|---|---|
| BIAS (%) | 132 | 130 | 739 | 72 |
| $F$ (%) | 39 | 43 | 640 | 5 |
| $H$ (%) | 92 | 86 | 99 | 67 |
| $C$ (%) | 66 | 60 | 13 | 64 |

the high TWL resulted in the failure of the artificial dunes in 7 cells of the domain (5 % of all the dunes). Even with the collapse of only 7 cells, the water was able to flow inland but with a more limited water supply, generating a flood pattern similar to the one of the simulations without protections and water depths 0.03 m smaller.

In terms of evaluation indices (Table 3), the E2015 shows a 132 % value of BIAS and 39 % value of $F$, due to the false alarm associated with some overestimation of the event. The simulation also exhibits a 92 % value of $H$ and 66 % value of $C$ showing a good representation of the flood extent even with misalignments between the flooded areas. Results for E2015D present similar pattern. Values for BIAS (130 %), $F$ (43 %), $H$ (88 %) and $C$ (59 %) indicates that both simulations can similarly reproduce the flood.

For the storm Denise of 2022, the simulation without protections (E2022) results in a maximum flood extent larger than the observed one, as the latter includes only areas in the proximity of the shoreline (Fig. 9a). In E2022 the flood pattern is like the event of 2015, with a maximum water level lower in the northern and southern portions of the domain (around 0.5 m) and higher for the central part of the study area (around 0.8 m). The highest value reached is 1.09 m. However, the realistic case with dunes (Fig. 9b)

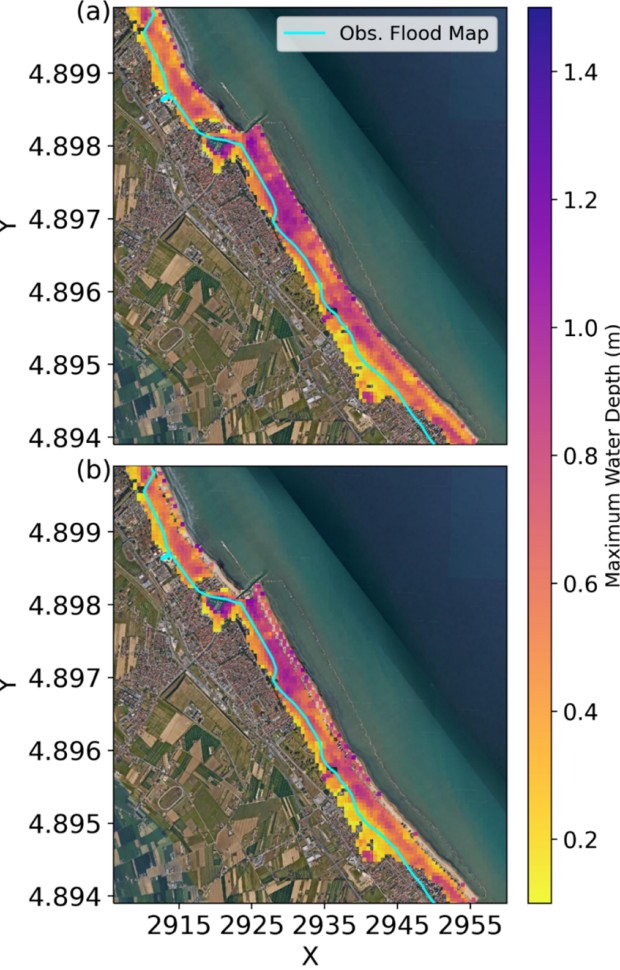

**Figure 8.** LISFLOOD-FP maximum water depth (m) for the 2015 simulation without protections E2015 **(a)** and with protections E2015D **(b)**. The cyan line corresponds to the limits of the observational flood area. ©Google Maps.

shows the much-reduced flood extent, consistent with observations (simulation E2022D). This time the dunes did not erode, and their protective action was evident.

The evaluation indices of E2022 display values of 640 % of $F$ and 739 % of BIAS due to the large overestimation in maximum flood extent (Table 3). The $H$ value of 99 % indicates that most of the cells identified as flooded in the observations are flooded also in the simulation, but the value of $C$ of 13 % indicates that the simulation results in many false positives. These values are much improved in the simulation with coastal protection (E2022D), where the flooded area broadly coincides with the observational one. In particular, the false alarm rate drops to 5 %, and the values of BIAS, $H$ and $C$ are reasonable, considering that the width of the flooded area is comparable with the resolution of the model.

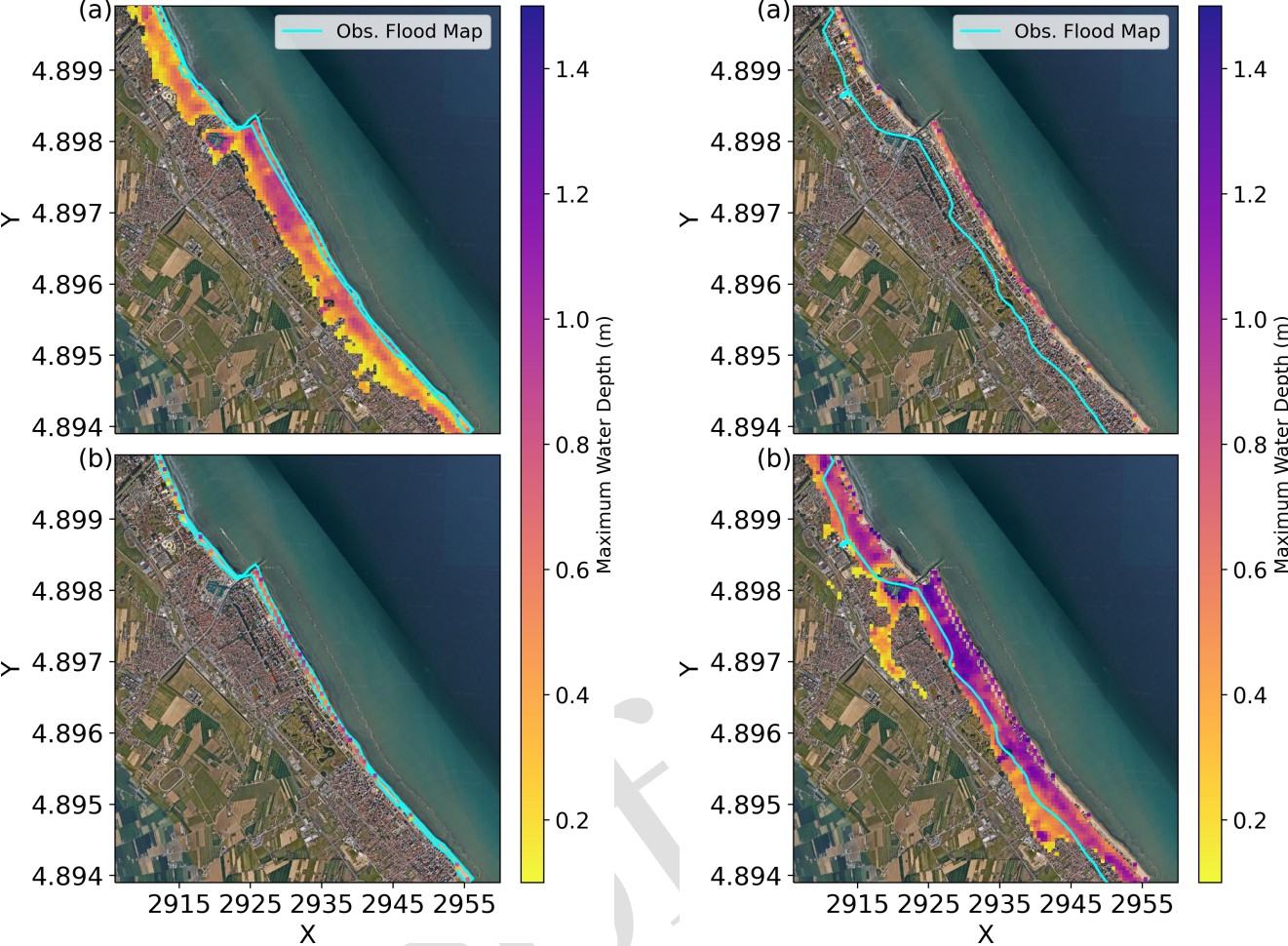

**Figure 9.** LISFLOOD-FP maximum water depth (m) for the 2022 simulation without protections E2022 **(a)** and with protections E2022D **(b)**. The cyan line corresponds to the limits of the observational flood area. ©Google Maps.

**Figure 10.** Waves' contribution to flood: simulation E2015DWL which does not consider the contribution of swash **(a)** and simulation E2015TWL which considers the full extent of TWL as contributing to both dune failure and water supply **(b)**. ©Google Maps.

## 3.2 Effects of swash on dune failure

As discussed in the previous section, during the 2015 event, dune failure occurred not preventing the large inundation. A correct representation of the waves' contribution is important for the event since dune failure depends on that. Simulation E2015DWL (Fig. 10a), which does not consider the contribution of swash, does not result in dune failure, and reproduces as inundated only tiny areas near the shoreline. By contrast, E2015TWL (Fig. 10b), which considers the full TWL as boundary condition overestimates the flood with a 18 % larger maximum flooded area compared with E2015D and is associated with water depths 0.3 m higher. Thus, we conclude that for the correct reproduction of the dune failure the contribution of the swash in TWL is important.

## 3.3 Effects of uncertainty

The uncertainty associated with the DTM (Fig. 11c) and wave conditions (Fig. 11b) exerts a significant influence on the extent of flooding observed in the simulations, whereas the uncertainty related to dune parameters is less pronounced for the 2015 event (Fig. 11a). In the UDUNE2015 simulations, discrepancies between the E2015 and E2015D scenarios are primarily confined to the cells of the portions of dunes that did not fail and areas with low water levels within the interior of the study domain (Fig. 11a). Variations in the DTM critically affect the structural integrity of the dunes under storm conditions, leading to a more pronounced impact of DTM uncertainty. Specifically, in the E2015D+ simulation, no dune failures were observed, while the E2015D− simulation exhibited failures in 28 cells, corresponding to 22 % of the dune structures. Consequently, the uncertainty area UDTM2015 is substantially larger compared to

UDUNE2015. Notably, the flooded area in the E2015D−simulation was 1315 % greater than that in E2015D+ and 7 % greater than E2015D. The highest level of uncertainty was observed in UWAVE2015, where the flooded area accounting for the full swash (E2015DTWL) was 1463 % larger than that using only the setup (E2015DWL). For both UWAVE2015 and UDTM2015, the largest discrepancies in flooded areas were directly linked to dune collapse. These findings highlight a strong nonlinearity in the relationship between flooded area and variations in the DTM and swash, with a critical threshold evident during the dune failure process.

For the 2022 event, the uncertainty associated with dunes (UDUNE2022; Fig. 12a) has the most significant impact on the extent of flooding observed in the simulations since they did not fail. Specifically, the flooded area in the E2022 scenario is 1317 % larger than that in E2022D. This is evident in Fig. 9, which illustrates that dune integrity is maintained, thereby confining flooding to the beach strip. In contrast, the uncertainties associated with swash contribution and the Digital Terrain Model (UWAVE2022 and UDTM2022; Fig. 12b) indicate no influence on dune collapses in this scenario. Consequently, the uncertainty for these parameters is negligible, with values effectively equal to zero.

The analysis of the uncertainties reveals that during the 2015 event, where dune collapse occurred, the largest source of uncertainty was associated with wave contributions, followed by uncertainties related to the Digital Terrain Model (DTM) and dune parameters. These findings suggest that the failure of even a small number of dunes can produce flooding conditions comparable to scenarios without dune protection, with the extent of flooding primarily influenced by water inflow and the regional topography. Conversely, for the 2022 event, in which the dunes withstood the storm, uncertainties related to the DTM, and swash contributions were insufficient to induce dune failure. This underscores the critical importance of dune integrity in determining the simulation outcomes.

## 4   Discussion

The results from simulation E2015D provide valuable insights into the flooding dynamics of 2015 and demonstrate that LISFLOOD-FP accurately reproduces inundation patterns for significant events by incorporating representations of dunes and swash dynamics. The interaction between these protections, waves, and water levels can be complex, as demonstrated by extreme events like the 2022 flood, when temporary dunes effectively safeguarded the shoreline, or the one of 2015, when they failed. The model successfully predicted dune failure during the 2015 event, while in 2022, the dunes effectively protected the land from inundation.

Our findings indicate that during the 2015 event, the swash significantly contributed to the erosion of these structures,

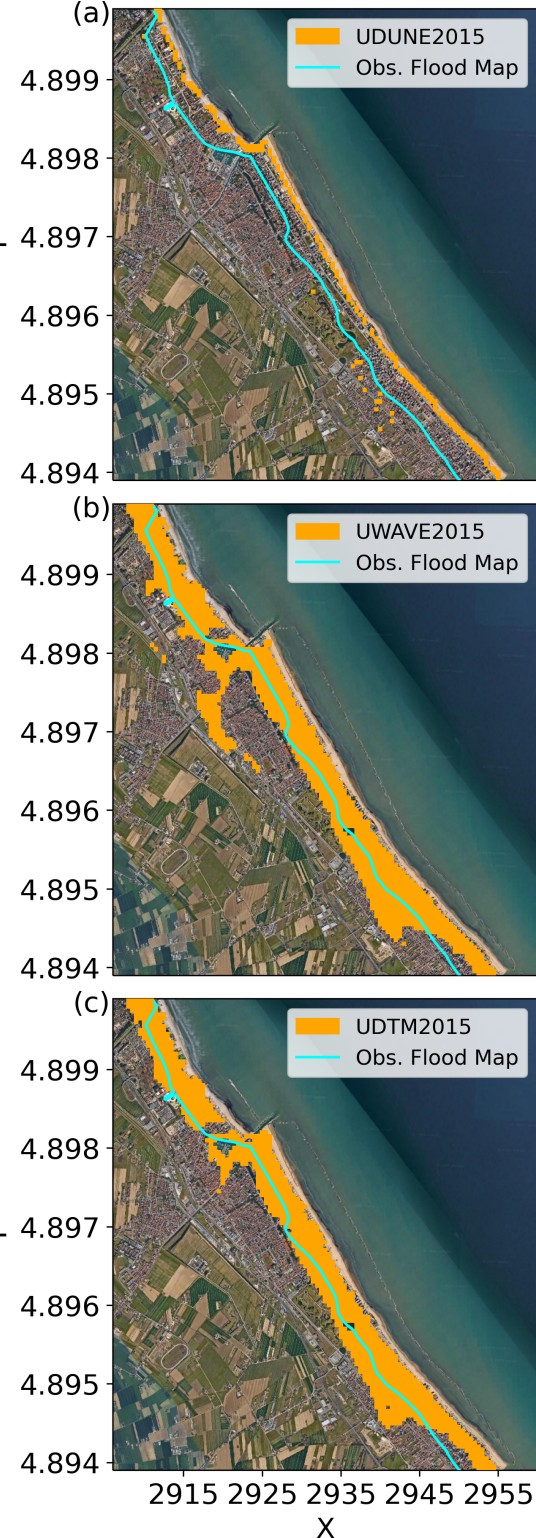

**Figure 11.** LISFLOOD-FP uncertainty associated with dunes UDUNE2015 (**a**), waves UWAVE2015 (**b**) and DTM UDTM2015 (**c**). Orange areas represent the uncertainty, given by the difference of flooded areas in the simulations. The cyan line corresponds to the limits of the observational flood map. ©Google Maps.

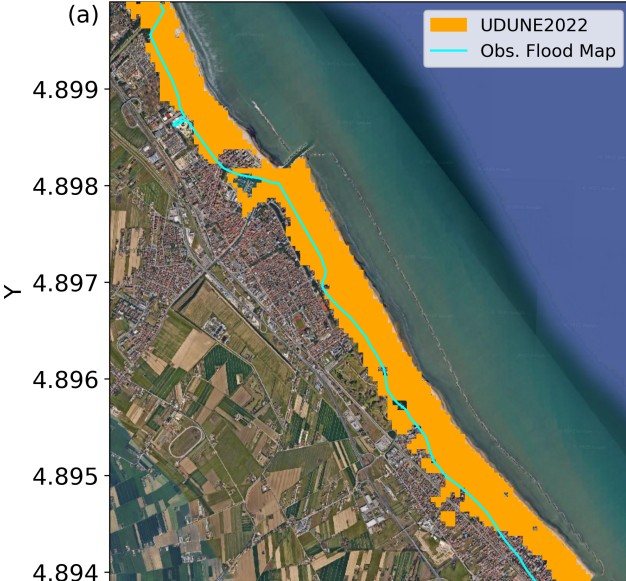

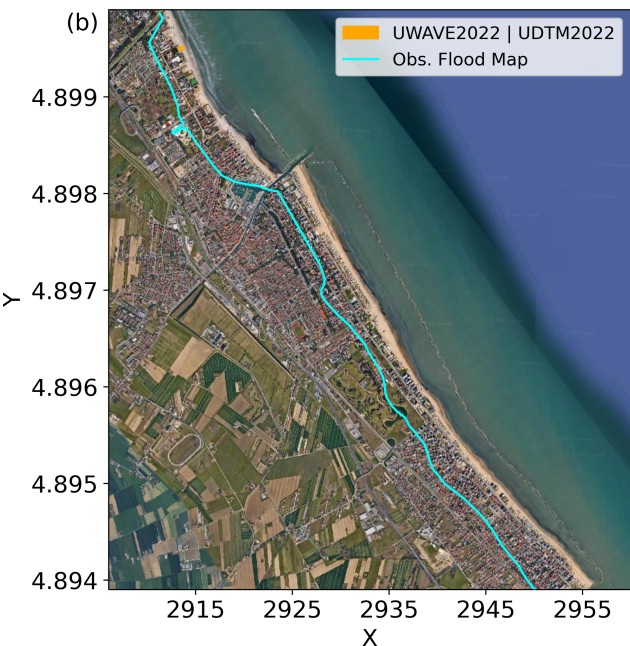

**Figure 12.** LISFLOOD-FP uncertainty associated with dunes UDUNE2022 **(a)**, waves UWAVE2022 and DTM UDTM2022 **(b)**. Orange areas represent the uncertainty, given by the difference of flooded areas in the simulations. The cyan line corresponds to the limits of the observational flood map. ©Google Maps.

reaching up to 0.4 m in areas where dune collapse occurred (Fig. 6d), ultimately allowing water to breach inland. Notably, even though only a small portion of the dunes (5 %) failed in our simulation, this was enough to trigger a flood as severe as if no protections had been in place. The sensitivity analysis of the model to the dune Failure Water Depth (FWD) showed that with a reduced height of 1.3 m, the

model presents, approximately 23 % of the dune cells failed, while for 1.2 m this increased to about 32 %. Conversely, simulations with higher FWD values resulted in no dune failures. These results highlight the strong influence of dune height on the modeled flooding dynamics.

Additionally, a possible outcome is that once a dune is breached, the remaining dunes may obstruct the floodwater's backflow, worsening the aftermath of the inundation. This effect was not clearly visible in the depth maps of the simulations presented in this study. However, additional simulations conducted under higher boundary conditions and with partial dune failure – where eroded sections were located adjacent to intact dunes – did show a tendency for increased maximum water depths landward of the non-eroded dunes. This suggests that residual dune segments can locally impede drainage and temporarily retain water, supporting the physical plausibility of the proposed mechanism. Although this behaviour was not systematically analyzed in the current work, it highlights an interesting hydrodynamic interaction that could be explored more explicitly in future studies through targeted simulations and higher-resolution topographic data.

The event of 2022, when the dunes successfully protected the coast, is characterized by significantly lower values of the swash than in 2015 (only 0.2 m, Fig. 7c). For this event, representing the dunes was critical for improving the model's accuracy, shifting the simulation bias from widespread overestimation of 739 % to a modest underestimation.

The precise representation of the dune structures and the corresponding flooded areas depends heavily on accurate DTM data and height measurements. Our uncertainty analysis, consistent with Dottori et al. (2022), shows that small changes in DTM data can significantly enlarge flooded areas. This is a crucial point, as demonstrated by the 2015 event, where the uncertainty in the DTM ranged from scenarios with no dune failure and minimal impact to a break scenario resulting in a significant flood and highlights that frequent topographic surveys are essential for effective flood forecasts in the context of disaster risk reduction and that temporal discrepancies between the surveys and the events introduce uncertainty in the initial conditions.

Dune failure, like any coastal protection failure, is inherently stochastic, governed by the interaction between structural characteristics and hydrodynamic forcing such as water levels and wave action. The evolution of dune erosion occurs across both time and space through processes including scarp formation, slumping, and sediment redistribution. These processes are strongly influenced by sedimentological properties – such as mineralogy, grain-size distribution, sorting, compaction, and biological content – which play a crucial role in determining dune resistance to storm impacts (Bertoni et al., 2014; Xie et al., 2020; De Falco et al., 2022).

An important limitation of the present modeling approach lies in its binary representation of dune failure, in which a dune cell is instantaneously and completely removed once

the total water level (TWL) exceeds the dune's Failure Water Depth (FWD). This simplification neglects the spatial and temporal complexity of dune erosion, meaning that a uniform FWD parameter may either overestimate or underestimate dune stability depending on local sedimentary and biological conditions. Nevertheless, this assumption represents a pragmatic compromise that enables coupling with a nonmorphodynamic model such as LISFLOOD-FP.

Despite its simplicity, the binary failure scheme provides a computationally efficient, first-order approximation that captures the hydrodynamic consequences of dune erosion and breaching. More sophisticated morphodynamical approaches, while physically more realistic, generally require extensive parameterisation and data inputs that are rarely available to coastal managers. The proposed binary framework thus provides a practical and parsimonious means of approximating floodplain dynamics with limited input requirements. Future developments of this approach will involve close collaboration with stakeholders to assess parameter availability and to explore the inclusion of partial or time-dependent erosion formulations, thereby enabling a more gradual and physically realistic representation of dune degradation while maintaining computational efficiency.

Finally, the implemented modeling framework is designed to allow flexibility in dune representation: dunes can be repositioned within the simulation domain and assigned varying FWD values. This capability enables the exploration of alternative dune configurations and failure scenarios, thereby improving the understanding of how dune position, continuity, and resistance influence coastal flooding dynamics, even under conditions of limited data availability.

Lateral boundary conditions at the coastline play a crucial role, particularly the inclusion of wave contributions for the TWL. In the E2015DWL simulation, which neglects the swash and uses only the wave setup, the flood is confined to the coast. Conversely, the E2015DTWL simulation, which accounts for the full swash contribution, extends the inundation further inland. We found that E2015D, using an overwash efficiency $\alpha = 0.25$, provided a satisfactory representation of the event. These findings highlight the importance of correctly representing the wave contribution to water supply: neglecting it leads to underestimation, while using TWL as the boundary condition leads to overestimation. It is important to underline, that in this study we set $\alpha = 0.25$ based on geometrical considerations and approximating the waves as triangular. But in general, the overwash efficiency $\alpha$ can be used as a calibration parameter to best fit the simulation results.

Our results align with the findings of Zhang and Najafi (2020) and Carneiro-Barros et al. (2023), emphasizing the critical interplay between various components of water levels. During the 2015 event (Fig. 6), the storm surge peak coincided with the peak of the waves, which were directed perpendicular to the shore. The occurrence of this event during neap tide, combined with the peak of residuals during low tide, suggests that the tide did not exacerbate the event's intensity. This implies that the impact would have been even more severe if had it occurred during spring tide. In contrast, the 2022 event (Fig. 7) was less impactful, despite residuals reaching levels comparable to the 2015 event. This was because both surge peaks coincided with low tide during spring tide, and the wave peak was not in phase with the storm surge, with mean wave directions not perpendicular to the shore. The occurrence of both events during low tide suggests that their severity could have been much greater, highlighting the critical need for continuous monitoring of dune conditions and timely forecasting to ensure a comprehensive risk management in Emilia Romagna.

The findings of this study also align with recent advancements in coastal flood modeling. Bertin et al. (2014) conducted a comprehensive analysis of coastal flood risk using a full hydrodynamic model. Their findings highlighted the model's remarkable ability to simulate coastal flood dynamics with high accuracy, emphasizing the critical role of detailed and precise data on the geometry of coastal defenses. However, the study did not consider the potential defense failures, which represents a significant limitation in understanding real-world flood risks. Additionally, while fully hydrodynamic models are praised for their precision and reliability, their applicability is constrained by substantial computational demands, which can hinder their use in large-scale or time-sensitive scenarios.

Closer to our approach, Leijnse et al. (2021) uses a shallow water equation model and incorporate a wave energy solver which translate offshore wave conditions into nearshore dynamics. However, our method bypasses the computational demands of a wave energy solver by directly integrating externally provided wave data. Geertsen et al. (2024) uses an intermediate complexity model and integrated it with an empirical dike failure model using conditional FWD levels. In contrast, our architecture is built inside the same code without the need to use separated models simplifying integration and facilitating alternative and more complex failure scenarios developments. Our enhanced model offers a streamlined, empirically grounded framework that maintains practical applicability without sacrificing detail.

The model's ability to represent the effects, failures, and drawbacks of coastal protection dunes, as well as quantify the contribution of waves, makes it a valuable tool for coastal hazard mapping. A possible way to overcome the limitation posed by the nonlinear nature of the uncertainty is using this LISFLOOD-FP in an ensemble framework. Additionally, this model can assist in defining appropriate failure heights for seasonal dunes in the region.

## 5 Conclusions

In this study, we enhanced the coastal flood modeling capabilities of LISFLOOD-FP by incorporating wave setup,

swash dynamics, and dune failure mechanisms under overwash conditions. Our validation against two contrasting storm events in Cesenatico demonstrates that accounting for these processes is essential for accurate coastal hazard mapping, particularly in regions like Emilia-Romagna where seasonal protective dunes are constructed annually.

The improved model introduces an overwash efficiency parameter to quantify swash contribution to flood volumes and enables dynamic DTM updates to capture dune erosion and topographic evolution during events. This approach bridges the gap between simplified floodplain models and computationally demanding morphodynamic simulations, providing a pragmatic tool for operational forecasting while maintaining physical realism in representing inflow pathways and inundation patterns.

A critical limitation emerges from uncertainty in dune geometry: variations of just a few centimeters in dune height can determine whether dunes survive or collapse, leading to non-linear propagation of uncertainty in simulated flood extent. The lack of detailed topographic surveys and observational flood maps compounds this challenge. An ensemble modeling approach, generating simulations across varied dune geometries and extreme sea level scenarios, could provide probabilistic hazard assessment and is computationally feasible given the model's efficiency. Complementing this, continuous monitoring of dune status using drone-based or fixed camera systems would reduce geometric uncertainty at reasonable cost and enable data-driven model updates.

This work represents a significant advancement toward coastal digital twins capable of supporting both prevention and response. The model enables optimization of coastal defence design by leveraging extreme event statistics and supports operational forecasting to guide protective actions during events. By providing computationally efficient yet physically grounded simulations, this approach offers a practical contribution to integrated coastal risk management.

*Data availability.* The Digital Terrain Model (DTM) used in this study was provided by the Geological, Soil and Seismic Area of the Emilia-Romagna region and is available through the regional geoportal https://geoportale.regione.emilia-romagna.it/catalogo/dati-cartografici/altimetria/layer-2 (last access: 31 December 2025). Tide gauge data were obtained from the Istituto Superiore per la Protezione e la Ricerca Ambientale (ISPRA) via the Italian Mareographic Network https://www.mareografico.it/ (last access: 31 December 2025). Wave data were provided by the Agenzia Regionale per la Prevenzione, Ambiente Energia dell'Emilia-Romagna (ARPAE) via https://simc.arpae.it/dext3r/ (last access: 31 December 2025). The modified LISFLOOD-FP model source code and the complete set of input data used for the simulated events are publicly available on Zenodo https://doi.org/10.5281/zenodo.18377755.

*Author contributions.* IRL conducted the conceptualization, software development, data elaboration and manuscript drafting. LM guided the supervision, conceptualization and manuscript drafting. LP supported with data provision, manuscript review, and revision. GC contributed for the funding, manuscript review and revision. IF, SC, MV, MM and NP contributed with the manuscript review and revision.

*Competing interests.* Maurilio Milella is employed by the company Environmental Surveys S.r.l. The remaining authors declare that the research was conducted in the absence of any commercial or financial relationships that could be construed as a potential conflict of interest.

ther geographical representation in this paper. The authors bear the ultimate responsibility for providing appropriate place names. Views expressed in the text are those of the authors and do not necessarily reflect the views of the publisher.

*Financial support.* This research has been supported by the European Space Agency (ESA) through the EOatSEE project, under the Earth Observation Science for Society block of activities, part of the FutureEO-1 programme (contract no. 4000138378/22/I-DT).

This research has also been supported by the the National Operational Programme for Research and Innovation 2014–2020 (CCI 2014IT16M2OP005), ESF REACT-EU resources, Action IV.4 "Doctorates and research contracts on innovation topics" and Action IV.5 "Doctorates on Green topics" (grant no. DOT199RYN9-6).

*Review statement.* This paper was edited by Piero Lionello and reviewed by Fabio Bozzeda and Giovanni Scardino.

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

**Remarks from the typesetter**

**TS2**     Authors statement: "These simulations were performed both with dunes (E2015D+, E2015D−, E2022D+, E2022D−) and without dunes (E2015+, E2015−, E2022+, E2022−), where the $\pm$ suffix indicates the addition or subtraction of the confidence value." should be replaced by: "These simulations were performed with dunes (E2015D+, E2015D−, E2022D+, E2022D−), where the $\pm$ suffix indicates the addition or subtraction of the confidence value." This change is required because the simulations without dunes (E2015+, E2015−, E2022+, E2022−) are neither presented nor discussed in the manuscript.