# Peer review of "Numerical modelling framework for assessing dune effectiveness against coastal inundation"

_EGUsphere, 2025_

## Author Comment (AC1)

**Dear Reviewer,**

We would like to sincerely thank you for your thorough and insightful review of our manuscript. Your detailed comments and constructive suggestions are greatly appreciated and have been invaluable in improving the quality and clarity of our work.

We are grateful for your positive assessment of the study's overall structure, methodological soundness, and relevance to coastal inundation modeling and coastal risk management. Your acknowledgment of the model's contribution toward more reliable and efficient forecasting tools, as well as its potential role in the development of coastal digital twins, is especially encouraging.

We have carefully considered each of your comments—both major and minor—and have revised the manuscript accordingly. Specifically, we have replied for the following points:

**Reviewer:**

1. Justification and Sensitivity of Key Parameters ( $\alpha$  and FWD)

The core of the methodological novelty lies in the introduction of two critical parameters: the overwash efficiency coefficient  $\alpha$  (eq. 2) and the dune Failure Water Depth (FWD). Their determination and impact on the results are of fundamental importance.

Overwash Efficiency ( $\alpha$ ): The authors set  $\alpha=0.25$  based on "geometrical considerations and approximating the waves as triangular" (lines 189-191). This justification is overly qualitative and requires more rigorous explication. What are these geometrical considerations precisely? It is suggested that the authors add a section or an appendix to illustrate the physical or geometrical reasoning behind this choice. More importantly, the manuscript would benefit immensely from a sensitivity analysis of this parameter. How do the results (inundation extent and dune failure for the 2015 event) change for values of  $\alpha$  ranging from 0 (setup only) to 1 (full swash contribution)? Such an analysis would not only strengthen the choice of 0.25 but also provide valuable insights into its role as a potential calibration parameter, as suggested by the authors themselves (lines 375-376)."

**Authors:**

We appreciate the reviewer's insightful comment and agree that the justification for setting the overwash efficiency ( $\alpha$ ) to 0.25 required a clearer and more physically grounded explanation explanation. The triangular wave shape used to define  $\alpha$  represents a good first approximation of the overwash process, as it captures the essential geometry of the uprush and backwash phases while remaining analytically simple. The mean water level beneath a symmetric triangular wave corresponds to half of its maximum height, and, since only the uprush portion contributes to overwash transport while the backwash returns seaward, the effective contributing fraction becomes approximately  $0.5 \times 0.5 = 0.25$ . Thus,  $\alpha = 0.25$

represents the fraction of the incident wave height that effectively drives landward overwash under this idealization.

In coastal environments, especially near the breaking and inner-surf zones, field and laboratory observations show that waves can reasonably be assumed as sawtooth-like shapes, characterized by a steep uprush face and a more gradual backwash (Suntoyo et al., 2008; Grasso et al., 2011; Bonneton, 2023). Such sawtooth waves deliver a similar uprush water volume as a symmetric triangular waveform because the shorter, steeper onshore face produces a rapid water-level rise over a reduced duration, effectively compensating for the longer but weaker backwash phase. When integrated over a wave period, the total uprush water flux remains of comparable magnitude to that of an equivalent triangular wave with the same height and period. Therefore, the geometric argument leading to  $\alpha = 0.25$  remains valid as a first-order representation of the overwash efficiency for both symmetric triangular and asymmetric saw-tooth waves.

Regarding the sensitivity analysis, while a site-specific calibration could refine the model performance, its results would depend strongly on local geomorphological and hydrodynamic conditions, require an extensive case-specific analysis, and, thus, be difficult to generalize beyond the study area. This supports the inclusion of both contributions through an intermediate efficiency value as a physically reasonable and transferable approximation. On the other hand,  $\alpha$  can be readily adjusted to reflect local hydrodynamic and morphological conditions, making the approach relocatable and adaptable to different coastal settings, a feature we considered worth mentioning in the manuscript.

The following explanation was added in the paper (lines 206-219):

"the overwash efficiency was set to  $\alpha=0.25$  which arises from a simple geometrical argument based on the idealized shape of a breaking wave. By approximating an individual wave as a triangular shape, the ratio of its mean height (representing the effective water volume contributing to overwash) to its maximum height is 0.5. However, only the uprush portion of the wave contributes to overwash transport, while the backwash is typically lost seaward. Assuming an approximately equal division between uprush and backwash, the effective fraction becomes 0.5 × 0.5 = 0.25. Hence,  $\alpha=0.25$  represents the fraction of the incident wave height that contributes effectively to overwash transport under this simplified geometrical assumption.

Although this argument is based on a symmetric triangular waveform, the same reasoning applies to asymmetric, saw-tooth-like waves characterized by a steep uprush and a more gradual backwash—waveforms commonly observed in the surf and swash zones and frequently adopted as first-order approximations in coastal engineering (Suntoyo et al., 2008; Grasso et al., 2011; Bonneton, 2023). In such cases, the geometric asymmetry alters the relative duration of the uprush and backwash phases but does not fundamentally change the proportional relationship between total wave height and the effective uprush volume contributing to

overwash. Therefore, the chosen value of  $\alpha$  = 0.25 remains a reasonable, physically consistent approximation for both symmetric and asymmetric (saw-tooth) wave shapes."

Bonneton P. Energy and dissipation spectra of waves propagating in the inner surf zone. Journal of Fluid Mechanics. 2023;977:A48. doi:10.1017/jfm.2023.878

Grasso, F., H. Michallet, and E. Barthélemy (2011), Sediment transport associated with morphological beach changes forced by irregular asymmetric, skewed waves, J. Geophys. Res., 116, C03020, doi:10.1029/2010JC006550.

Suntoyo, Tanaka, H., Sana, A., 2008. Characteristics of turbulent boundary layers over a rough bed under saw-tooth waves and its application to sediment transport. Coastal Engineering 55, 1102–1112.

**Reviewer:**

Failure Threshold (FWD): The dunes were modeled with a uniform FWD of 1.4 meters (line 186). This is a strong assumption. How was this value determined? Is it based on literature specific to the artificial dunes of the Emilia-Romagna region, on post-event observations, or on engineering criteria? Is its uniformity along the entire coastline realistic? As the success or failure of the dunes is the centerpiece of the analysis, a detailed justification for this value is imperative. Here too, a sensitivity analysis, at least for the 2015 event, showing how the number of failed dune cells varies with the FWD (e.g., 1.3 m, 1.4 m, 1.5 m), would be extremely insightful and would demonstrate the model's robustness (or sensitivity) to this parameter."

**Authors:**

We appreciate the reviewer's comment and agree that our explanation of the dune representation lacked important details. Unfortunately, detailed structural information was not available to us after interaction with the regional authorities. Therefore, in absence of more detailed information, we mapped the dune based on imagery of 2015, and looked for a configuration of the dunes that would provide a reasonable estimation for both 2015 and 2022. The following information was added to section 2.3 (lines 194-203):

"For the simulations in Cesenatico (ER), a mapping of the seasonal dunes in the area was carried out using high-resolution Google satellite imagery acquired in March 2015. Since these images were taken after the storm event, only the locations where dunes withstand the storm or had been reformed could be clearly identified. The mapping focused on delineating the spatial position of dune crests through visual interpretation of the dune ridges. Only the geolocation points were incorporated into the model, while dune geometry (width) was constrained by the 50 m grid resolution. As detailed information on dune morphology for 2015 and 2022 was unavailable, we sought a configuration capable of accurately reproducing both events. The dune

height was assigned a uniform value corresponding to a Failure Water Depth (FWD) of 1.4 m to all grid cells, based on a sensitivity analysis. However, the model structure allows the assignment of different FWD or dune height values for each grid cell, enabling future applications to incorporate spatial variability when more detailed morphological data become available."

We also agree that a more detailed sensitivity analysis of the FWD is warranted. The discussion is somehow made when we discussed the UDTM where we added/removed 0.07 m not to the dunes but for the whole DTM. In lines x-x we present the consequences of changing the DTM for the 2015 event and no difference for the 2022 event:

"Variations in the DTM critically affect the structural integrity of the dunes under storm conditions, leading to a more pronounced impact of DTM uncertainty. Specifically, in the E2015D+ simulation, no dune failures were observed, while the E2015D- simulation exhibited failures in 28 cells, corresponding to 22% of the dune structures."

Furthermore, we made additional simulations varying the FWD between 1.2 and 1.5 m for both events. In 2022's scenarios, it did not significantly affect the results. For 2015, if we decrease the FWD, the ratio of failing dunes increases with minimal implications on the inundation extent. Conversely, an increase in FWD to the point where no dune fails, present an underestimation of the event. The following information was added to the discussion (lines 369-372):

"The sensitivity analysis of the model to the dune Failure Water Depth (FWD) showed that with a reduced height of 1.3 m, the model presents, approximately 23% of the dune cells failed, while for 1.2 m this increased to about 32%. Conversely, simulations with higher FWD values resulted in no dune failures. These results highlight the strong influence of dune height on the modeled flooding dynamics."

**Reviewer:**

2. Simplification of the Dune Failure Mechanism

The implemented failure model is binary in nature: when TWL > FWD, the dune is "entirely" and instantaneously removed from the terrain (line 138). While this is an understandable and pragmatic simplification for a non-morphodynamic model like LISFLOOD-FP, its implications must be discussed more thoroughly. Dune erosion is a progressive process in both time and space.

o It is suggested that a paragraph be added to the Discussion (Section 4) that explicitly acknowledges this limitation. Instantaneous failure could, for instance, overestimate the peak water discharge into the hinterland compared to a more gradual erosion process. How might this simplification affect the inundation hydrograph and the maximum flood extent? o Furthermore, the model removes the entire dune cell. What occurs if TWL only slightly exceeds FWD? Is it realistic for the entire dune (50m wide, according to the model resolution) to be eroded instantaneously? The discussion should contextualize this approach as a first, yet effective, step toward modeling these complex processes, perhaps mentioning potential future developments (e.g., partial or time-parametrized erosion).

**Authors:**

We thank the reviewer for this insightful comment regarding the simplification of the dune failure mechanism. We fully agree that representing dune erosion as a binary, instantaneous process is a pragmatic but idealized approach. Specifically, we note that instantaneous dune removal could potentially overestimate or underestimate dune stability. This simplification may therefore affect the shape of the inundation hydrograph and the maximum flood extent, particularly in areas where the dune is only marginally overtopped. However, it is important to note that the Failure Water Depth (FWD) is not necessarily equivalent to the dune crest elevation or total dune height. Instead, FWD represents a threshold water level above which the structural integrity of the dune is assumed to fail, leading to a breach of the barrier. This parameter is thus conceptual and can be used to more realistic represent the combined effects of dune geometry, sediment characteristics, and antecedent conditions, rather than representing a purely geometric property.

Similarly, the model grid resolution (50 m) should not be interpreted as the physical width of an individual dune. The removal of a dune cell when the TWL exceeds the FWD represents the functional loss of the dune's protective capacity within that cell, rather than the wholesale physical removal of a 50 m-wide dune.

We have clarified these conceptual distinctions in the revised manuscript and explicitly state that this binary representation should be regarded as a first, computationally efficient approximation to capture the hydrodynamic consequences of dune failure. As noted in the Discussion, this framework provides a foundation upon which more gradual, time-dependent, or morphodynamically coupled erosion schemes can be developed in future work respecting data limitations and stakeholders' requirements. The following clarifying paragraphs were added (lines 391–414):

"Dune failure, like any coastal protection failure, is inherently stochastic, governed by the interaction between structural characteristics and hydrodynamic forcing such as water levels and wave action. The evolution of dune erosion occurs across both time and space through processes including scarp formation, slumping, and sediment redistribution. These processes are strongly influenced by sedimentological properties—such as mineralogy, grain-size distribution, sorting, compaction, and biological content—which play a crucial role in determining dune resistance to storm impacts (Bertoni et al., 2014; De Falco et al., 2022; Xie et al., 2020).

An important limitation of the present modeling approach lies in its binary representation of dune failure, in which a dune cell is instantaneously and completely removed once the total water level (TWL) exceeds the dune's Failure Water Depth (FWD). This simplification neglects the spatial and temporal complexity of dune erosion, meaning that a uniform FWD parameter may either overestimate or underestimate dune stability depending on local sedimentary and biological conditions. Nevertheless, this assumption represents a pragmatic compromise that enables coupling with a non-morphodynamic model such as LISFLOOD-FP.

Despite its simplicity, the binary failure scheme provides a computationally efficient, first-order approximation that captures the hydrodynamic consequences of dune erosion and breaching. More sophisticated morphodynamical approaches, while physically more realistic, generally require extensive parameterisation and data inputs that are rarely available to coastal managers. The proposed binary framework thus provides a practical and parsimonious means of approximating floodplain dynamics with limited input requirements. Future developments of this approach will involve close collaboration with stakeholders to assess parameter availability and to explore the inclusion of partial or time-dependent erosion formulations, thereby enabling a more gradual and physically realistic representation of dune degradation while maintaining computational efficiency.

Finally, the implemented modeling framework is designed to allow flexibility in dune representation: dunes can be repositioned within the simulation domain and assigned varying FWD values. This capability enables the exploration of alternative dune configurations and failure scenarios, thereby improving the understanding of how dune position, continuity, and resistance influence coastal flooding dynamics, even under conditions of limited data availability."

**Reviewer:**

3. Details on Dune and DTM Mapping

The reproducibility of the study is critically dependent on the quality of the input data, particularly the topography. The authors state that they mapped the seasonal dunes from "satellite imagery" (line 185) and incorporated them into the DTM. More details are required:

What were the source and resolution of the satellite imagery?

- What methodology was used to extract the dune geometry (height, width, exact position) from the images? Was this a manual or semi-automated process?
- What are the resolution and acquisition date of the base DTM provided by the "Geological, Seismic and Soil Service of the ER region"? Its age relative to the studied events could be relevant.

These details are fundamental, as the uncertainty analysis (Section 3.3) excellently demonstrates how sensitive the results are to small variations in the DTM (±7 cm).

**Authors:**

We thank the reviewer for this comment and acknowledge that our description of the dune and DTM mapping was previously incomplete. For the dunes, detailed structural data are not available; therefore, we used satellite imagery to determine the dune's spatial extent, adjusted the width to correspond with the model grid resolution, and estimated the dune height based on consultations with local authorities and sensitivity analyses. The following information has been added to Section 2.3 (lines 194-203):

"For the simulations in Cesenatico (ER), a mapping of the seasonal dunes in the area was carried out using high-resolution Google satellite imagery acquired in March 2015. Since these images were taken after the storm event, only the locations where dunes withstand the storm or had been reformed could be clearly identified. The mapping focused on delineating the spatial position of dune crests through visual interpretation of the dune ridges. Only the geolocation points were incorporated into the model, while dune geometry (width) was constrained by the 50 m grid resolution. As detailed information on dune morphology for 2015 and 2022 was unavailable, we sought a configuration capable of accurately reproducing both events. The dune height was assigned a uniform value corresponding to a Failure Water Depth (FWD) of 1.4 m to all grid cells, based on a sensitivity analysis. However, the model structure allows the assignment of different FWD or dune height values for each grid cell, enabling future applications to incorporate spatial variability when more detailed morphological data become available."

For the DTM, we have included comprehensive information on the spatial resolution, coordinate reference system, and acquisition date, in order to ensure full transparency and reproducibility of the dataset. The following information has been added to Section 2.3 (lines 189–193):

"The DTM was provided by the Geological, Seismic and Soil Service of the ER region with a spatial resolution of 5 m, referenced to the WGS84/UTM Zone 32N coordinate system (EPSG:32632) and an acquisition date of 2009. A coastline mapping was carried out to provide boundary points coordinates in the sea/land interface. The coastline is determined by analyzing the DTM, identifying the zero-crossing, and designating the first positive point as its location. The model was set on a domain covering the area of Cesenatico with a resolution of 50 m. The resulting grid has a size of 150x121 grid cells."

**Reviewers:**

4. Presence of Placeholder Text

On page 18, at the beginning of section 3.2, there is an entire paragraph of Latin placeholder text ("Suspendisse a elit ut leo pharetra cursus..."). cancel please."

**Authors:**

We thank the reviewer for pointing this out. The presence of placeholder text in Section 3.2 was an oversight on our part. This has now been corrected in the revised manuscript.

**Minor Comments**

**Reviewer:**

• Lines 72-74 (Digital Twin): The introduction mentions the "digital twin" concept. While pertinent, the connection could be made more explicit. The proposed framework is a component of a potential digital twin, not a complete one (which would imply a continuous data stream and assimilation). It might be more accurate to frame it as a "fundamental step toward the development of a digital twin for coastal management."

**Authors:**

Changed "critical step toward developing a digital twin for sustainable coastal management" to "fundamental step toward the development of a digital twin for coastal management."

**Reviewer:**

• Lines 190-191 (Beach-face slope): The value of  $\beta f = 0.05\%$  appears extremely low (1:2000). The reference to Ciavola et al. (2006) should be double-checked, as such a gentle slope is atypical for the foreshore. Was 5% or 0.05 (dimensionless) intended? Please check and clarify.

**Authors:**

We thank the reviewer for pointing this out. Indeed, the value was incorrectly written as 0.05 %. The correct beach-face slope is  $\theta f = 5$  %, as originally intended. This has been corrected in the revised manuscript.

**Reviewer:**

• Lines 354-355 (Backflow): In the discussion, it is hypothesized that the remaining dunes may obstruct the backflow of water. This is an excellent and physically plausible observation. Did the model actually exhibit this behavior? If so, it would

be useful to mention this explicitly, perhaps by indicating areas of "ponding" behind the non-failed dunes on the depth maps. If it is only a hypothesis not directly supported by the results, it should be phrased as such.

**Authors:**

In the shown simulations, this behaviour was not explicitly observed. However, additional simulations conducted with higher boundary conditions did exhibit water ponding behind the remaining dunes, supporting the physical plausibility of this mechanism. The following paragraph was added to the discussion (lines 374-380):

"This effect was not clearly visible in the depth maps of the simulations presented in this study. However, additional simulations conducted under higher boundary conditions and with partial dune failure—where eroded sections were located adjacent to intact dunes—did show a tendency for increased maximum water depths landward of the non-eroded dunes. This suggests that residual dune segments can locally impede drainage and temporarily retain water, supporting the physical plausibility of the proposed mechanism. Although this behavior was not systematically analyzed in the current work, it highlights an interesting hydrodynamic interaction that could be explored more explicitly in future studies through targeted simulations and higher-resolution topographic data."

---

## Author Comment (AC2)

Dear Reviewer,

We would like to sincerely thank you for your thoughtful and constructive review of our manuscript. We greatly appreciate your positive assessment of our work and your recognition of its relevance to understanding dune effects in coastal inundation modeling. Your detailed line-by-line suggestions have been extremely valuable in improving the clarity, accuracy, and overall readability of the manuscript.

Your comments have been carefully addressed in the revised version. Specifically, we have:

Reviewer:

Line 32. Hereinafter, you have to report "Sea-Level Rise".

**Authors:**

**Corrected Sea Level to Sea-Level**

Reviewer:

Line 33. Add also the reference for IPCC 2021.

**Authors:**

**Added the reference for IPCC 2021.**

Reviewer:

Line 60. Change form in landform.

**Authors:**

**Changed form in landform**

Reviewer:

Line 61. Change beach interface in backshore.

**Authors:**

**Changed to beach interface in backshore.**

Reviewer:

Lines 61-62. Here, you refer to storm surge and wave overwash as causes of dune erosion. However, you also reported dune breaching in the abstract, which is one of the most evident effects of dune erosion during a storm. Please revise the list of storm-related effects to include dune breaching.

**Authors:**

Thank you for this valuable comment. We acknowledge that dune breaching is indeed one of the most evident and severe outcomes of dune erosion during storm events. The storm surge and wave overwash will act as primary drivers of both dune erosion and dune breaching. In response to your suggestion, we have revised the introduction to include a more detailed explanation of the dune erosion process, highlighting how these mechanisms can lead to breaching under extreme conditions (lines 64-72).

"Coastal dune erosion refers to the landward retreat of sandy beaches and dune systems as a result of storm-induced wave action and elevated water levels. The extent of this erosion can be described using an erosion hazard scale (Leaman et al., 2021) based on the degree of horizontal recession experienced during a storm. At the lowest level, minor beach narrowing occurs when the beach width is reduced but the dune system remains unaffected. As erosion intensifies, substantial beach narrowing takes place, where the dune system is still intact but becomes more vulnerable to damage from subsequent storms. More severe conditions lead to dune face erosion, in which erosion progresses landward from the dune toe but does not yet reach the crest. Under the most extreme circumstances, dune retreat occurs, where significant erosion impacts and undermines the landward side of the dune crest, leading to a loss of dune volume and a reduction in the coastal buffer that protects inland areas from storm surges and flooding."

Reviewer:

Line 67. Substitute "are erected" with "have been built".

**Authors:**

Substituted "are erected" with "have been built".

Reviewer:

Lines 97-98. Revise the definition of wave runup, which is not a contribution to the TWL. The wave runup is defined as the maximum vertical extent to which a high-energy wave reaches the coastal landforms above the instantaneous water level (e.g. Villarroel-Lamb and Williams, 2022).

**Authors:**

We thank the reviewer for this valuable observation and for providing the reference. We acknowledge that, strictly speaking, wave runup represents the maximum vertical extent of wave uprush above the instantaneous water level. However, in coastal engineering, TWL usually defines the sum of tide, surge, and wave runup (Carneiro-Barros et al.,2025; Hsu et al.,2023; Stockdon et al.,2023). In our study, the

concept of wave runup is useful to evaluate the interaction between waves and coastal dunes. Nevertheless, when estimating the water volume available for inundation, we recognize that the full extent of the swash should not be included. For this reason, we introduced the concept of a supply total water level (STWL), which accounts for the water level contribution relevant to inundation processes without incorporating the entire runup excursion.

Carneiro-Barros, Jose Eduardo, Ajab Gul Majidi, Theocharis Plomaritis, Tiago Fazeres-Ferradosa, Paulo Rosa-Santos, and Francisco Taveira-Pinto. 2025. "Coastal Flooding Hazards in Northern Portugal: A Practical Large-Scale Evaluation of Total Water Levels and Swash Regimes" Water 17, no. 10: 1478. https://doi.org/10.3390/w17101478

Hsu, C.-E., Serafin, K. A., Yu, X., Hegermiller, C. A., Warner, J. C., and Olabarrieta, M.: Total water levels along the South Atlantic Bight during three along-shelf propagating tropical cyclones: relative contributions of storm surge and wave runup, Nat. Hazards Earth Syst. Sci., 23, 3895–3912, https://doi.org/10.5194/nhess-23-3895-2023, 2023.

Stockdon, H.F., Long, J.W., Palmsten, M.L. et al. Operational forecasts of wavedriven water levels and coastal hazards for US Gulf and Atlantic coasts. Commun Earth Environ 4, 169 (2023). https://doi.org/10.1038/s43247-023-00817-2

Reviewer:

Line 135. Change in "The approach proposed in this study is based on the work of ....."

**Authors:**

Changed in "The approach proposed in this study is based on the work of ....." to "The approach proposed in this study draws inspiration from Shustikova et al. (2020), who developed a methodology for the representation of levees and their breaching processes."

Reviewer:

Line 160. Insert some toponyms in Figure 3 and a scale bar.

**Authors:**

We thank the reviewer for this helpful suggestion. We have added relevant toponyms and a scale bar to Figure 3. This improvement has indeed enhanced the clarity and interpretability of the figure.

Figure 1: Emilia-Romagna's coast in the northeast of Italy. Yellow dots represent Porto Corsini's tide gauge (north) and Nausicaa's wave buoy (south). Red rectangle represents the modeled area in the town of Cesenatico. © Google Maps

**Reviewer:**

Line 180. When you describe the DTM features, provide also some info about the Reference System, in particular about the orthometric elevations (which is the datum?). This is very important if you are applying a simulation based on tide gauge and buoy data.

**Authors:**

We appreciate the reviewer's observation and agree that including information about the reference system is essential. We have therefore added details about the reference system used for the DTM, including the orthometric elevation datum, to ensure clarity and consistency in the description of our data sources and their

application within the modeling framework. The following was added in lines 189-190:

"The DTM was provided by the Geological, Seismic and Soil Service of the ER region with a spatial resolution of 5 m, referenced to the WGS84/UTM Zone 32N coordinate system (EPSG:32632) and an acquisition date of 2009."

**Reviewer:**

Line 205. Maybe Figure 6 and Figure 7 can be merged into a unique figure, because boundary conditions for the two storm events are reported in a similar way.

**Authors:**

We appreciate the reviewer's suggestion. However, after several attempts to merge the figures, we found that each contains a substantial amount of information and combining them would negatively affect the overall layout and readability of the manuscript and make the boundary condition patterns more difficult to interpret. Therefore, we have chosen to keep Figures 6 and 7 separate to preserve clarity and facilitate comparison between the two storm events.

**Reviewer:**

Lines 289-291. There are some unreadable sentences.

**Authors:**

The presence of placeholder text in Section 3.2 was an oversight on our part. This has now been corrected in the revised manuscript.

**Reviewer:**

Line 365. Several important factors influencing dune erosion are not addressed in the Discussion section, namely sediment mineralogy, grain size, and biological factors. In particular, it is necessary to cite references stating that dune nourishment requires compatible sediments. If the nourishment sediments are incompatible with the native material in terms of mineralogy, grain size, and biological components, the dune will fail to act as an effective barrier against storm impacts and will be more susceptible to erosion.

**Authors:**

We sincerely thank the reviewer for this insightful comment. We fully agree that sediment mineralogy, grain size, and biological factors play a crucial role in dune erosion and nourishment performance. Following the reviewer's suggestion, we have expanded the discussion section to address these aspects in more detail and have included relevant references highlighting the importance of sediment

compatibility in effective dune nourishment. This addition has strengthened the discussion and improved the overall completeness of the manuscript. The following paragraph was added to the discussion (line 391-414):

"Dune failure, like any coastal protection failure, is inherently stochastic, governed by the interaction between structural characteristics and hydrodynamic forcing such as water levels and wave action. The evolution of dune erosion occurs across both time and space through processes including scarp formation, slumping, and sediment redistribution. These processes are strongly influenced by sedimentological properties—such as mineralogy, grain-size distribution, sorting, compaction, and biological content—which play a crucial role in determining dune resistance to storm impacts (Bertoni et al., 2014; De Falco et al., 2022; Xie et al., 2020).

An important limitation of the present modeling approach lies in its binary representation of dune failure, in which a dune cell is instantaneously and completely removed once the total water level (TWL) exceeds the dune's Failure Water Depth (FWD). This simplification neglects the spatial and temporal complexity of dune erosion, meaning that a uniform FWD parameter may either overestimate or underestimate dune stability depending on local sedimentary and biological conditions. Nevertheless, this assumption represents a pragmatic compromise that enables coupling with a non-morphodynamic model such as LISFLOOD-FP.

Despite its simplicity, the binary failure scheme provides a computationally efficient, first-order approximation that captures the hydrodynamic consequences of dune erosion and breaching. More sophisticated morphodynamical approaches, while physically more realistic, generally require extensive parameterisation and data inputs that are rarely available to coastal managers. The proposed binary framework thus provides a practical and parsimonious means of approximating floodplain dynamics with limited input requirements. Future developments of this approach will involve close collaboration with stakeholders to assess parameter availability and to explore the inclusion of partial or time-dependent erosion formulations, thereby enabling a more gradual and physically realistic representation of dune degradation while maintaining computational efficiency.

Finally, the implemented modeling framework is designed to allow flexibility in dune representation: dunes can be repositioned within the simulation domain and assigned varying FWD values. This capability enables the exploration of alternative dune configurations and failure scenarios, thereby improving the understanding of how dune position, continuity, and resistance influence coastal flooding dynamics, even under conditions of limited data availability."